# Domain Adaptation for Conversational Query Production with the RAG Model Feedback

**Ante Wang**[1,2]**, Linfeng Song**[3]**, Ge Xu**[4]**, Jinsong Su**[1,2][*]

[1]School of Informatics, Xiamen University, China
[2]Key Laboratory of Digital Protection and Intelligent Processing of Intangible Cultural Heritage of Fujian and Taiwan (Xiamen University), Ministry of Culture and Tourism, China
[3]Tencent AI Lab, Bellevue, WA
[4]College of Computer and Control Engineering, Minjiang University, China
wangante@stu.xmu.edu.cn, jssu@xmu.edu.cn

## Abstract

Conversational query production is an emerging fundamental task for the dialogue system, where search queries are generated to explore the vast and continually updating knowledge from a search engine. To accelerate this line of research, previous studies have released several datasets with human-annotated search queries. However, the limited annotations still can not cover conversations of various domains. To solve this challenge, we propose a novel domain adaptation framework. It is inspired by a weakly supervised learning algorithm from previous work (Wang et al., 2023b) that guides a model using reinforcement learning with BM25 scores as feedback. Though effective, it is fragile facing noisy content on webpages from a commercial search engine and variance in conversations because of ignoring deep semantic information of dialogue contexts. Thus, we improve the algorithm by taking the advance of retrieval-augmented generation (RAG) and exploring several practical techniques such as knowledge distillation for stable training. We conduct experiments in multiple settings across different languages. Guided by the RAG model feedback, our model is more robust and performs significantly better especially in a more challenging setting over strong baselines.[1]

## 1 Introduction

Leveraging external knowledge has been proven to be important for various text-generation tasks (Lewis et al., 2020b). Very recently, with the burgeoning of large language models, it can effectively alleviate the hallucination issue and improve faithfulness (Nakano et al., 2021; Glaese et al., 2022). Along this line, exploring the Internet for external knowledge is gaining popularity due to its continually updated content and broad coverage on a variety of domains (Komeili et al., 2022). To access this type of knowledge, a model is required

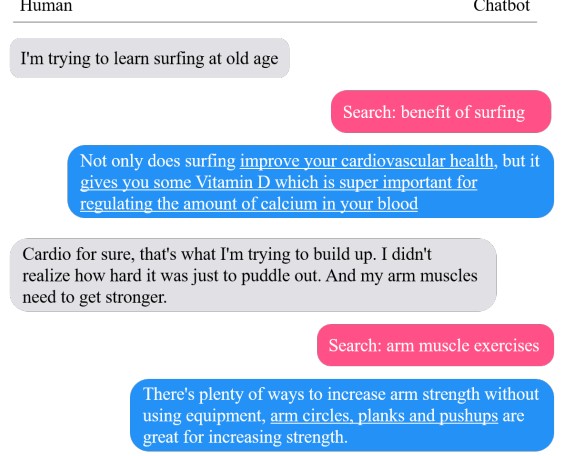

Figure 1: A conversation with 4 dialogue turns, where the chatbot sends search queries to a search engine for retrieving external knowledge (not shown) before generating the responses. With the aid of the search engine, generated responses can be more engaging and helpful (underlined).

to generate search queries for interacting with a search engine. In this work, we focus on *conversational query production*. It aims to generate search queries reflecting user intents so that better responses can be obtained with the help of external knowledge retrieved from a search engine. A typical example is shown in Figure 1. This task is conceptually similar to other knowledge-intense tasks such as Question Answering (Nakano et al., 2021). However, it is more challenging because a conversational query producer has to mine user intents from complex dialogue contexts involving multiple utterances.

Naturally, the current efforts consider query production as a text-to-text generation task and can be generally sorted into two categories: *supervised* and *unsupervised* learning. In this first category, previous studies (Komeili et al., 2022; Zhou et al., 2022) are dedicated to collecting conversations, where the query for each utterance is annotated. With the development of pretrained language mod-

---

[*]Corresponding author.
[1]https://github.com/DeepLearnXMU/DAMF.

els ([Raffel et al., 2020](); [Lewis et al., 2020a]()), a strong query producer can be obtained with only thousands of training instances. However, these models may still be suboptimal in an out-of-domain conversation. As another category, [Wang et al. (2023b)]() propose to train their query producer with a weak-supervised learning algorithm free of costly human efforts. They first extract keywords from the dialogue context as candidate search queries, which are then fed to *Wikipedia search*[2] for corresponding documents. Next, the quality of each candidate is evaluated by comparing its retrieved documents against the gold response using the BM25 algorithm ([Robertson and Walker, 1994]()). Finally, the query producer is trained to select the best keywords as search queries by maximizing corresponding BM25 scores (rewards) using reinforcement learning.

Inspired by these studies, we try to tackle the domain adaptation challenge by combining both worlds. As a tough combination, we first train a model on the labeled source-domain instances in the standard supervised training manner. Then, the model is adapted to the target domain based on the weak-supervised learning algorithm introduced above. Though working effectively, we notice that there exist two under-explored issues, severely preventing our model from further improving.

First, a commercial search engine always returns noisy webpages containing unrelated information such as advertisements. However, [Wang et al. (2023b)]() only study in a toy setting by using Wikipedia search to simulate a search engine, where the returned Wikipedia pages are much cleaner. Another problem attributes to the variation of conversations. [Wang et al. (2023b)]() conduct experiments on an impractical scenario, where nearly all utterances involve external knowledge. Nevertheless, there often exist variances that some utterances can be generated without any external knowledge (e.g., greetings). As there are no corresponding queries to these instances, training on them will inevitably hurt the model performance. Concretely, the above two problems result in inaccurate BM25 scores and unstable reinforcement learning, challenging the robustness of the adaptation algorithm.

To tackle these problems, we first take advantage of retrieval-augmented generation (RAG, [Lewis et al. 2020b]()) and replace the BM25 algorithm with a trained RAG model. Compared with the BM25 algorithm only considering the surface word overlap, the RAG model can better capture the deep semantic information from the dialogue context and learn to evaluate each document based on its contribution to the target response. Then, we heuristically filter some potentially harmful instances where candidate queries are scored too close or lower than a given threshold, indicating that these queries are indistinguishable or low-quality. Lastly, the standard REINFORCE algorithm ([Williams, 1992]()) adopted in ([Wang et al., 2023b]()) is fragile when facing such noisy rewards. This makes the trained query producer always too far from the good initial policy that is trained on the well-labeled source-domain dataset. Therefore, we further introduce a regularization objective based on knowledge distillation ([Hinton et al., 2015]()), avoiding involving too large gradient updates (often leading to a bad policy) away from the initial policy.

We conduct experiments on two domain adaptation settings: CLEAN and NOISY, according to whether a model will be influenced by the two issues described above. In the CLEAN setting, we carry out experiments on Wizard-of-Internet ([Komeili et al., 2022]()) → Wizard-of-Wikipedia ([Dinan et al.]()). We follow [Wang et al. (2023b)]() to use Wikipedia search as the search engine. In the NOISY setting, we set up DuSinc ([Zhou et al., 2022]()) → KdConv ([Zhou et al., 2020]()) and DuSinc → DuConv ([Wu et al., 2019]()) where *Sogou search*[3] is adopted, which is a Chinese commercial search engine. Experiment results show that our model significantly outperforms strong baselines, especially in the NOISY setting. Further analysis demonstrates that our designed model feedback is more accurate than BM25 scores and all proposed operations can benefit the final model.

## 2 Background

In this section, we first introduce how a dialogue system equipped with a query producer works. Formally, taking a concatenated dialogue history $\mathcal{X}_{<t} = u_1, ..., u_{t-1}$ of $t-1$ turns as inputs, a query producer generates a search query $q$. Then, $q$ will be sent to a search engine for retrieving a list of $N_A$ documents $\mathcal{K}^q = k_1^q, ..., k_{N_A}^q$. Finally, a response $u_t$ is generated by a response generator consuming both $\mathcal{X}_{<t}$ and $\mathcal{K}^q$. In this work, we mainly focus on the crucial query producer.

---

[2] https://www.wikipedia.org/

[3] https://www.sogou.com/

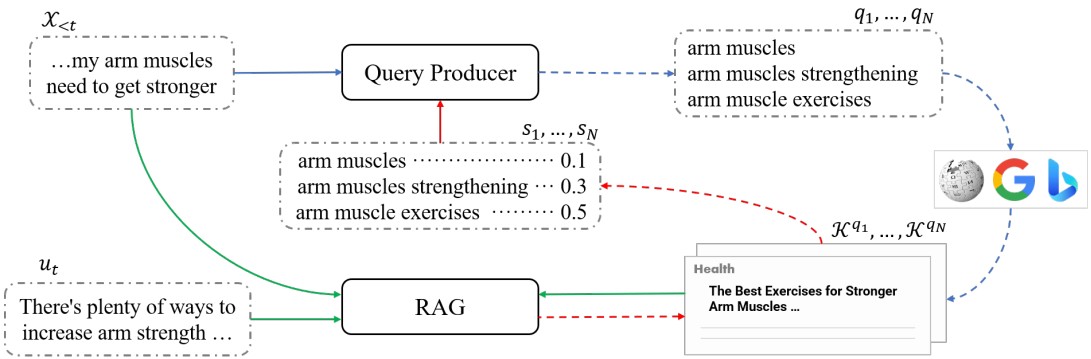

Figure 2: The adaptation workflow for a query producer well-trained in the source domain, where solid lines (→) and dashed lines (⇢) indicate training and inference passes. At **stage 1** (→), the query producer consumes input utterances $\mathcal{X}_{<t}$ to get a set of search queries $q_1, ..., q_N$. Then (⇢), we retrieve relevant documents $\mathcal{K}^{q_1}, ..., \mathcal{K}^{q_N}$ for these queries from a search engine. At **stage 2** (→), a RAG model is trained by maximizing the probability of dialogue response $u_t$ given both $\mathcal{X}_{<t}$ and $\mathcal{K}^{q_1}, ..., \mathcal{K}^{q_N}$. During this process, the retriever from the RAG model learns the contributing score of each document to response generation. At **stage 3** (⇢), supervision signals $s_1, ..., s_N$ are calculated based on retrieval scores from the last step to update (→) the query producer.

Following the common practice (Komeili et al., 2022), the query producer can be initialized with a T5 model (Raffel et al., 2020) that is pretrained on a massive text corpus. On an annotated query generation benchmark, the query producer is then trained to generate the target query $q$ in an autoregressive manner. Formally, this model takes the standard cross-entropy (CE) loss as the training objective:

$$\mathcal{L}_{\text{CE}} = -\log p(q \mid \mathcal{X}_{<t}; \theta), \tag{1}$$

where $\theta$ represents the model parameters.

## 3 Domain Adaptation Framework

Figure 2 visualizes the workflow of our framework. Before adaptation, the query producer has been initialized with a source-domain model ($\theta_{src}$), which is trained using $\mathcal{L}_{\text{CE}}$. Intuitively, the model can predict some good but not perfect queries for the target domain because of the domain shift.

To tackle this issue, we then adopt our framework which can be decomposed into 3 stages for better clarification. At stage 1 (§3.1), we create an offline cache for all possible queries (generated by the source-domain model) and their retrieved documents from a search engine for target-domain instances. They will be used to train the models in later stages. At stage 2 (§3.2), we train the RAG model on the target-domain dataset using documents from the offline cache. This model will play the role of the reward model for reinforcement learning in stage 3. At stage 3 (§3.3), the query producer ($\theta_{src}$) is transferred to the target domain using reinforcement learning, where the RAG

model scores for the queries in the offline cache are calculated as rewards. With the informative model feedback, our framework can be more effective than traditional algorithms such as *Self-training* (Kulshreshtha et al., 2021).

### 3.1 Offline Cache Creation

Since querying a search engine during the model training is time-consuming, we follow Wang et al. (2023b) to set up an offline cache storing documents for all possible queries before the actual model training. However, a query producer will take a large search space of candidate queries, which is exponential to the average query length. Thus, we only consider at most $N$ queries with high probabilities for each instance, which can be easily accessed via *beam search* (Sutskever et al., 2014)[4].

Optionally, inspired by (Wang et al., 2023b), the candidate query set can be enriched with keywords extracted from the dialogue context using some public keyphrase extractors, e.g., TagMe (Ferragina and Scaiella, 2010). This is because the target query sometimes is a keyword that the model-predicted query set does not cover due to the limitation of the model capability in the target domain.

### 3.2 Training a RAG Model

With the conversations and related documents from the offline cache, we now are able to train a RAG

---

[4]We also explore *diverse sampling* (Fan et al., 2018; Holtzman et al.), but it does not make a difference in our preliminary study possibly due to the limited length of a query (mostly less than 5 words).

model. RAG is first proposed in (Lewis et al., 2020b) for various knowledge-intensive generation tasks. It consists of a *retriever* and a *generator*. During inference, the retriever will obtain several documents, each of which is treated as a latent variable and fed to the generator for predicting the target response. Then, the predictions of the generator given different documents are marginalized for the final prediction.

Usually, the retriever is initialized from the Dense Passage Retriever (DPR, Karpukhin et al. 2020), which is built upon Wikipedia and only serves in English. To satisfy wider use, we simplify the model by sharing the parameters of the retriever and generator encoder. Similar to the query producer, we initialize our implemented RAG model with T5. Formally, given the dialogue context $\mathcal{X}_{<t}$, the response $u_t$ and corresponding retrieved documents $\mathcal{K} = \mathcal{K}^{q_1}, ..., \mathcal{K}^{q_N}$ for this instance from the offline cache, we define the predicted probability to $u_t$ as

$$p(u_t \mid \mathcal{X}_{<t}) \approx \prod_i^{|u_t|} \sum_j^k p(z_j)p(u_{t,i} \mid \mathcal{X}_{<t}, u_{t,<i}, z_j; \theta_{gen}),$$
$$z_1, ..., z_k = \text{top-k}(p(\cdot \mid \mathcal{X}_{<t}, \mathcal{K}; \theta_{ret})),$$
(2)

where top-k($*$) is a function selecting $k$ documents (e.g., $z_j$) with the highest retrieval scores fed to the generator for predicting and $p(z_j) = p(z_j \mid \mathcal{X}_{<t}, \mathcal{K}; \theta_{ret})$. $\theta_{ret}$ and $\theta_{gen}$ denote parameters for the retriever and the generator, respectively.

Concretely, the retriever consists of the shared T5 encoder (T5-enc) and a randomly initialized Multilayer Perceptron (MLP) head. It consumes the concatenation of $\mathcal{X}_{<t}$ and each document $k_i \in \mathcal{K}$, and then produces the probability distribution over $\mathcal{K}$, indicating their importance for predicting $u_t$:

$$p(\cdot \mid \mathcal{X}_{<t}, \mathcal{K}; \eta) = \text{Softmax}(e_1, ..., e_{|\mathcal{K}|}),$$
$$e_i = \text{MLP}(\text{T5-enc}([\text{CLS}], \mathcal{X}_{<t}, k_i)[0]),$$
(3)

where [CLS] is a special token prepended for producing the vector fed to the MLP head.

One drawback of our RAG model is the slower retrieving process compared with the traditional one. This is because our model adopts a cross-encoder structure that takes concatenated dialogue context and document pairs as inputs. While previous work uses a bi-encoder structure, which separately encodes each part and calculates scores using Maximum Inner Product Search (MIPS). However,

the cross-encoder structure allows for richer interactions between the input context and a candidate document (Humeau et al.). It also saves computation costs by directly feeding the intermediate hidden states of the shared encoder to the generator decoder for predicting $u_t$.

Note that our implementation is generally based on RAG-Token (Lewis et al., 2020b), which jointly considers multiple documents during predictions. It echoes the previous study (Shuster et al., 2021) that RAG-Token is a better choice for this task than RAG-Sequence, which produces an output based on exactly one document.

### 3.3 Reinforcement Learning with the RAG Model Feedback

We use reinforcement learning to finetune the query producer, where output scores over candidate documents of the trained RAG model from the last stage are used as feedback.

**Scoring function based on retrieval scores** Formally, for a search query $q$ and its retrieved document list $\mathcal{K}^q$ with $N_A$ documents, the query score $s$ is decided by a scoring function $f(\mathcal{K}^q, *)$, where $*$ can be some optional inputs (e.g., $\mathcal{X}_{<t}$ and $u_t$). Wang et al. (2023b) define $f$ as

$$f(\mathcal{K}^q, u_t) = \max(\{\text{BM25}(k_i^q, u_t)\}_{\forall k_i^q \in \mathcal{K}^q}). \quad (4)$$

As the BM25 algorithm is based on surface word overlap, its scores may be inaccurate without taking deep semantic information into account. In this work, we define

$$f(\mathcal{K}^q, \mathcal{X}_{<t}, \eta) = \max(\{e_1^q, ..., e_{N_A}^q\}), \quad (5)$$

where $e_i^q$ is the retriever ($\theta_{ret}$) output given $k_i^q$ from $\mathcal{K}^q$ (Equation 3). In this way, we can easily obtain the scores for all possible queries of each instance from the offline cache.

**Quality control via filtering** Because of the noisy webpages and variance in conversations, it is risky to train the model on all instances without quality control. We propose two easy-to-implement operations to clean the dataset. For an instance with query scores $s_1, ..., s_N$ and $s_{max}$ / $s_{min}$ denoting the maximum / minimum within them, we only keep the instance where $s_{max} > \alpha$ and $s_{max} - s_{min} > \beta$, $\alpha$ and $\beta$ are hyperparameters.

Using the first operation ($\alpha$-*Filtering*), we aim to filter the instances where no external knowledge is required. For these instances, the candidate query

scores are generally low as their retrieved documents can not benefit response generation. Via the latter operation ($\beta$-*Filtering*), we drop the instances where their query scores are quite similar, indicating that they are difficult to differentiate by the reward model and thus are less helpful for our query producer.

**Reinforcement finetuning with regularization** We adopt the REINFORCE algorithm (Williams, 1992) to finetune the query producer using the query scores as the rewards. For a training instance, we have obtained a set of queries $q_1, ..., q_N$ and their scores $s_1, ..., s_N$ via the above process. To reduce the training variance, we define the reward $r_i$ for $q_i$ by normalizing the query scores $r_i = \frac{s_i - s_{min}}{s_{max} - s_{min}}$. Besides, we subtract a baseline value $\bar{r}$ by averaging the rewards. Formally, the reinforcement learning loss is defined as

$$\mathcal{L}_{rl} = -\Delta(\tilde{r}, \bar{r}) \log p(\tilde{q} \mid \mathcal{X}_{<t}; \theta_{tgt}), \quad (6)$$

where $\tilde{q}$ is sampled from the model predictive probability distribution over the candidate query set, and $\tilde{r}$ is the corresponding reward value.

A common issue for domain adaptation is hurting the model performance in the source domain. Besides, though we control the quality of training instances by filtering operations, some unrecognized harmful instances still make the training unstable. Considering that we have obtained a query producer $\theta_{src}$ fitted to a well-constructed source-domain dataset, we adopt this model to regularize the training process via knowledge distillation (KD, Hinton et al. 2015). Note that we also perform a filtering operation by dropping the queries with their confidence lower than a different threshold $\gamma$ to construct a small high-quality KD corpus. For each instance from the KD corpus, we ask the model to predict the pseudo query $\hat{q}$ generated by $\theta_{src}$:

$$\mathcal{L}_{kd} = -\log p(\hat{q} \mid \mathcal{X}_{<t}; \theta_{tgt}). \quad (7)$$

Thus the overall training loss for adaptation is

$$\mathcal{L} = \mathcal{L}_{rl} + \lambda \mathcal{L}_{kd}, \quad (8)$$

where $\lambda$ is a hyperparameter used to balance the two loss terms.

## 4 Experiment

### 4.1 Setup

**Dataset** We prepare 3 domain adaptation benchmarks involving 5 datasets across 2 languages for

| System | R@1 | R@3 | R@5 |
|---|---|---|---|
| T5-base | 54.78 | 69.07 | 73.51 |
| QP-Ext† | 62.41 | 72.91 | 74.87 |
| QP-Gen† | 56.77 | 66.08 | 68.22 |
| WSMF w/ BM25 | 65.91 | 77.69 | 80.48 |
| WSMF | 66.20 | 76.77 | 79.27 |
| Self-Train | 52.31 | 65.97 | 70.50 |
| Self-Train (MIX) | 52.76 | 66.39 | 70.86 |
| DAMF w/ BM25 | 67.63 | **78.49** | **81.02** |
| DAMF (ours) | **68.34** | 78.32 | 80.96 |

Table 1: Main results in CLEAN setting (WoI → WoW), where † denotes the results from (Wang et al., 2023b).

evaluation. Both **CLEAN** and **NOISY** settings are investigated. We mainly focus on the latter one, which is more challenging.

We take Wizard-of-Internet (WoI, Komeili et al. 2022) → Wizard-of-Wikipedia (WoW, Dinan et al.) as the CLEAN setting. Following Wang et al. (2023b), we adopt *Wikipedia search* and take the target Wikipedia page titles provided in the dataset for evaluation. As the Wikipedia page is well-constructed and nearly all instances require external knowledge, it is easier to train a query producer. The NOISY setting includes DuSinc (Zhou et al., 2022) → KdConv (Zhou et al., 2020) and DuSinc → DuConv (Wu et al., 2019). We manually annotate 817 and 321 queries for KdConv and DuConv as test sets, respectively. We use *Sogou search*, which is a Chinese commercial search engine. Thus, the returned documents always contain unrelated information. Besides, both KdConv and DuConv include many instances that need not search queries, nearly covering 32% and 45% according to our annotations.

Detailed introductions to these datasets and annotation guidelines are provided in Appendix A and B.

**Evaluation Metrics** We report the average results of 3 runs for all experiments. We mainly focus on query production, but also evaluate response generation to validate the positive effects of improving query quality.

- **Query production** For WoW, we follow Wang et al. (2023b) to use recall, denoted as R@$K$ ($K \in \{1, 3, 5\}$), which compares retrieved documents of the top $K$ predicted queries with ground-truth documents to evaluate the performance of query producers. For KdConv and DuConv, we follow previous work (Zhou et al., 2022) to use Unigram F1 (Uni. F1), BLEU-1/2

| System | KdConv | | | DuConv | | |
|---|---|---|---|---|---|---|
| | Uni. F1 | BLEU-1/2 | ROUGE-1/2/L | Uni. F1 | BLEU-1/2 | ROUGE-1/2/L |
| T5-base | 60.21 | 57.57/54.72 | 70.95/59.11/70.08 | 64.18 | 54.90/52.36 | 70.55/61.95/69.53 |
| text-davinci-003 | 54.93 | 49.38/45.29 | 67.53/54.23/66.24 | 61.44 | 57.62/55.79 | 69.82/60.16/68.95 |
| WSMF w/ BM25 | 53.98 | 41.60/41.05 | 61.71/50.96/61.54 | 50.97 | 30.82/30.54 | 56.56/46.94/56.26 |
| WSMF | 60.54 | 45.87/45.41 | 65.85/57.18/65.70 | 58.18 | 37.43/37.23 | 62.54/54.52/62.44 |
| Self-Train | 62.21 | 60.31/57.54 | 72.39/61.17/71.68 | 67.20 | 58.51/56.06 | 72.67/64.94/71.40 |
| Self-Train (MIX) | 62.42 | 60.31/57.54 | 72.71/61.47/71.93 | 68.08 | 60.10/57.73 | 73.45/66.04/72.17 |
| DAMF w/ BM25 | 64.27 | 63.71/60.10 | 73.40/62.05/72.27 | 68.82 | 61.34/58.80 | 74.52/66.96/73.18 |
| DAMF (ours) | **67.44** | **65.98/62.81** | **75.66/66.29/74.79** | **72.32** | **62.27/59.99** | **76.29/70.12/75.18** |

Table 2: Main results in the NOISY setting (DuSinc → KdConv & DuSinc → DuConv).

(Post, 2018) and also introduce ROUGE-1/2/L (Lin, 2004) for comparing generated queries against the annotated ones. Among these metrics, BLEU-1/2 mainly considers precision, while the others are more comprehensive by jointly considering precision and recall.

- **Response generation** As implemented in (Dinan et al.), we use Perplexity (PPL) and Uni. F1 to evaluate the predicted responses against gold references.

## 4.2 Main Results

Table 1 and 2 show the main test results on query production in CLEAN and NOISY settings. We list our models and some typical baselines from previous efforts: (1) *T5-base*. It is trained on the source-domain dataset using the CE loss (Eq. 1) and directly tested on the target-domain datasets. (2) *text-davinci-003*[5]. It is a strong large language model from the popular GPT3 family (Brown et al., 2020). We follow a common practice (Zhao et al., 2021) using 8-shot in-context learning to evaluate its query production performance. An input case is provided in Appendix C. (3) *QP-Ext* and *QP-Gen*. Both models are trained in the weakly supervised algorithm (Wang et al., 2023b). The former is fine-tuned from ELECTRA-base (Clark et al., 2020), aiming at extracting keywords from the context as queries. The latter is based on BART-base, generating queries using typical sequence generation. (4) *WSMF* and *WSMF w/ BM25*. They are our implemented *QP-Gen* by optionally replacing its BM25 scores with RAG retrieval scores. (5) *Self-Train* and *Self-Train (MIX)*. Self-training is a popular technique for domain-transfer learning. We follow Kulshreshtha et al. (2021) and He et al. to implement these models, where *Self-Train* only adopts

model-predicted pseudo generations as targets and *Self-Train (MIX)* additionally uses gold queries in the source domain. (6) *DAMF* and *DAMF w/ BM25*. They are trained using our framework. Again, we provide *DAMF w/ BM25* for a fair comparison, which uses BM25 as the base of $f$. Our model configuration is shown in Appendix D.

For the CLEAN setting (Table 1), we observe that: (1) Weakly supervised learning approaches even give better performance than *T5-base*. It is partially because the queries can usually be extracted from the contexts for WoW. Besides, *QP-Ext* and *QP-Gen* give inferior results than our implemented *WSMF w/ BM25* because we adopt a stronger pretrained model (T5-base). (2) For domain adaptation approaches, our framework can help train stronger models, while self-training even hurt model performance. It is because the source-domain model can not generate good pseudo targets due to the large domain discrepancy and this leads to severe error propagation. In comparison, our models are better guided by pseudo targets with their quality scores decided by $f$. (3) Using different $f$ gives competitive results, with *DAMF / WSMF* slightly better than *DAMF w/ BM25 / WSMF w/ BM25* on the main metric R@1. However, we still notice that RAG feedback is better supervision given our later analysis (§4.4). Thus, we believe that the advantage is not obvious due to the ideal scenario of the CLEAN setting and the robustness of the framework.

For the challenging NOISY setting (Table 2), we can draw the following conclusions: (1) *text-davinci-003* performs well using only 8-shot demonstrations, but is still worse than the *T5-base* finetuned on full source-domain instances. (2) Comparing *WSMF* and *WSMF w/ BM25*, the former performs significantly better thanks to the much more accurate model feedback, which is

[5]This model is tested by calling the official OpenAI api https://platform.openai.com/docs/models.

| System | Uni. F1 | BLEU-1/2 | ROUGE-1/2/L |
|---|---|---|---|
| DAMF w/o $\mathcal{L}_{kd}$ | 66.89 (-2.36) | **66.21**/62.75 (**-0.39**/-0.71) | 75.04/65.66/74.13 (-1.99/-2.32/-1.82) |
|   w/ BM25 | 63.12 | 60.82/56.79 | 73.40/62.05/72.27 |
|   w/ keywords | 66.81 | 62.60/60.53 | 73.21/65.11/72.63 |
|   w/o $\alpha$-Filtering | 66.08 | 65.07/61.60 | 74.80/64.90/73.81 |
|   w/o $\beta$-Filtering | 66.58 | 65.82/62.32 | 74.84/65.29/73.97 |
|   w/ $\mathcal{L}_{kd}$ | **67.44** (-0.56) | 65.98/**62.81** (-0.90/**-0.67**) | 75.66/66.29/74.79 (-0.90/-0.57/-0.72) |
|   w/ $\mathcal{L}_{skd}$ | 66.84 (-1.48) | 65.86/62.67 (-0.87/-0.90) | 75.21/65.71/74.34 (-1.25/-1.46/-1.32) |
|   w/ $\mathcal{L}_{CE}$ | 66.48 (-1.02) | 65.44/62.30 (-1.44/-1.57) | 74.78/65.25/74.01 (-0.97/-1.13/-0.92) |
| DAMF w/o $\mathcal{L}_{rl}$ | 65.49 | 63.62/60.70 | 74.55/64.50/73.73 |

Table 3: Ablation Study on DuSinc → KdConv. To study the negative impacts on the source domain, we also provide performance decreases on the source-domain data (DuSinc) which are shown in brackets.

a strong proof validating our motivation. Both of these models give low BLEU scores, it is because they usually only generate the main keywords without other details due to the unsupervised setup and thus have a large length penalty of BLEU metrics.[6] (3) Again, though *Self-Train* and *Self-Train (MIX)* give better results than *T5-base* this time, our model still significantly outperforms these baselines. Besides, *DAMF* significantly surpasses *DAMF w/ BM25* in this challenging NOISY setting, convincingly demonstrating the effectiveness of our framework.

### 4.3 Ablation Study

Table 3 shows the ablation study for our framework. It is conducted on DuSinc → KdConv, which is challenging and annotated more queries, thus can provide more convincing results. We can draw the following conclusions. **First**, we again observe that using RAG feedback works better than adopting BM25 scores, when $\mathcal{L}_{kd}$ is not implemented. We also explore enriching the query set with keywords appearing in the dialogue context as Wang et al. (2023b) suggested.[7] However, it instead hurts the results of BLEU and ROUGE, which is because some unrelated keywords may disturb the model training. Besides, both filtering operations can slightly improve the final results and save some training costs by dropping some harmful instances. **Second**, we compare some choices of regularization loss terms. $\mathcal{L}_{skd}$ is a variant of $\mathcal{L}_{kd}$, where the Cosine Similarity loss term is adopted to close the predicted distribution of source-domain and target-domain models on the small KD corpus.[8]

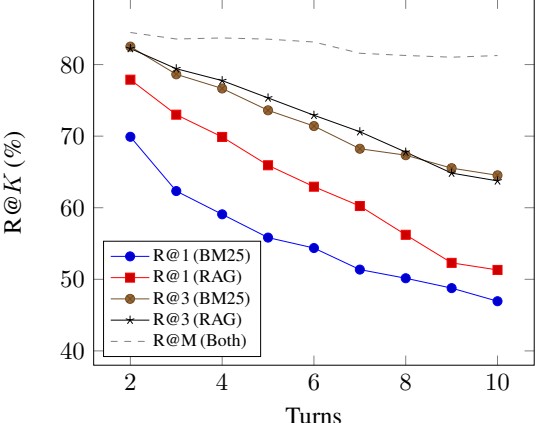

Figure 3: Evaluation of query scores calculated with different scoring functions ($f$) at various dialogue turns on WoW training set, where R@M is the ceiling performance.

However, it gives inferior results to $\mathcal{L}_{kd}$ on most metrics. It is because the source-domain model can be unconfident on target-domain instances, leading to unreasonable probability distributions. Another choice is jointly training with instances from the source domain ($\mathcal{L}_{CE}$). However, it performs worst in the target domain and is also less effective than $\mathcal{L}_{kd}$ for regularization. **Finally**, we remove the $\mathcal{L}_{rl}$ thus only keep the $\mathcal{L}_{kd}$ term for finetuning. It also helps but lags behind our model, validating the effectiveness of reinforcement learning. Note that it is similar to *Self-Train*, but it is trained on our small KD corpus, verifying the effectiveness of using RAG model feedback from another aspect.

### 4.4 RAG Model Feedback is More Accurate

The selection of $f$ decides the accuracy of the query scores, directly influencing the effectiveness of vital reinforcement learning in stage 3. Thus, we provide a comparison of $f$ using BM25 or RAG scores in Figure 3, validating that the RAG scores can be more accurate than BM25 scores. Thanks to

---

[6]For example, these models prefer "Lionel Messi" when the user asks "the birthplace of Lionel Messi".

[7]We use an open tool, TexSmart (https://ai.tencent.com/ailab/nlp/texsmart/zh/index.html), to extract entities as keywords.

[8]Details are introduced in Appendix E.

| System | WoW | | KdConv | |
|---|---|---|---|---|
| | PPL↓ | Uni. F1 | PPL↓ | Uni. F1 |
| Seq2seq | 17.52 | 22.08 | 10.48 | 33.92 |
| T5-base | 15.97 | 22.68 | 9.91 | 37.60 |
| Rank-Gen† | 21.60 | 17.81 | – | – |
| Merge-Gen† | 20.20 | 18.15 | – | – |
| WSMF w/ BM25 | 15.53 | 22.96 | 10.50 | 36.39 |
| WSMF | 15.47 | 23.06 | 10.28 | 36.41 |
| Self-Train (MIX) | 15.98 | 22.59 | 9.82 | 38.02 |
| DAMF w/ BM25 | 15.51 | 22.91 | 9.57 | 38.36 |
| DAMF (ours) | **15.38** | **23.11** | **9.27** | **38.99** |

Table 4: Main test results for response generation, where † denotes the results from (Wang et al., 2023b).

the gold document annotations (page titles) from WoW, we can directly evaluate $f$ using R@$K$, representing the hitting of the gold document with top-$K$ scored queries. Comparing the main metric R@1, using RAG feedback is always better than using BM25 and the conclusion holds across all dialogue turns. But they give similar results on R@3. It shows that the RAG model mainly helps select the best queries. Besides, we observe that both $f$ perform worse when the conversation continues. However, the ceiling performance only drops slightly. It is due to the much higher complexity of a longer context, challenging the understanding capacity of the models. Another performance boost can be expected when using stronger RAG variants (Shuster et al., 2021; Paranjape et al.). We leave it as future work.

We also provide a case from KdConv to further validate our conclusions, which could be found in Appendix F.

### 4.5 Response Generation

Table 4 shows the response generation results, where the models are equipped with corresponding query producers. All our models adopt the RAG model introduced in §3.2, except *Seq2seq*, which does not consider external knowledge inputs. Additionally, we provide *Rank-Gen* and *Merge-Gen*, which are the two types of models proposed in Wang et al. (2023b) equipped with *QP-Ext*. Generally, the performance of response generation grows when feeding external documents retrieved with better search queries. This validates the importance of developing stronger query producers. Some may concern that the improvement of response generation is limited even though there is a large margin for query production. It is because sometimes the target knowledge can be covered by a non-perfect

query. For example, when the user asks about "the birthplace of Lionel Messi", the target knowledge can also be retrieved by sending "Lionel Messi" to a search engine. However, we believe the query "Lionel Messi" is not good, as it does not faithfully reflect the user intent.

## 5 Related Work

### 5.1 Knowledge-aided Dialogue Model

In recent years, effective dialogue agents have explored various knowledge for facilitating dialogue understanding and response generation, such as sentiment (Song et al., 2022), speaker emotion (Poria et al., 2019), discourse structure (Wang et al., 2021a, 2023a), and external knowledge from different knowledge sources. This work falls into the last category of the above research lines, which has shown effectiveness in alleviating the hallucination problem. Early studies (Sun et al., 2019; Zhou et al., 2020; Dinan et al.; Wang et al., 2021b) explored static databases such as built knowledge graphs or collected documents (e.g., a Wikipedia dump). Besides, as a typical dialogue model has to select correct knowledge and generate conversation simultaneously, some researchers also explore fusing the QA and dialogue systems together (Adolphs et al., 2021; Oh et al., 2023). Recently, to build stronger models especially based on large language models (LLMs), leveraging vast knowledge from the Internet is gaining popularity because of its dynamically updating nature (Komeili et al., 2022; Zhou et al., 2022).

### 5.2 Conversational Search Query Production

Leveraging vast knowledge from the Internet is an important topic for developing various NLP tasks. We focus on conversational query production, which is the key to building a search-engine-aided dialogue system. Though it is recently proposed, this field has attracted wide attention. Following common practice, Komeili et al. (2022) and Zhou et al. (2022) have proposed Wizard-of-Internet and DuSinc with collections of conversations annotated with search queries for English and Chinese respectively to train their query producers. Wang et al. (2023b) proposed a weakly supervised learning algorithm for building a query producer free of human annotations.

In this work, we extend these studies to serve conversations with different distributions by proposing a novel domain adaptation framework.

We notice that (Chen et al., 2022) shares some same spirits with our approach. They study several reinforcement learning rewards for enhancing the question answering performance by improving a question rewriting model, which works similarly to our query producer. However, their methods may not be well adapted to conversation query production because of the task discrepancy. Besides, their rewards either suffer from the noisy issues we have discussed or are not interpretable compared with our used retrieval scores.

Our work is also remotely related to building large language models accessible to a search engine for knowledge-intensive tasks, such as WebGPT (Nakano et al., 2021) and Sparrow (Glaese et al., 2022). However, both of them developed their systems by adopting reinforcement learning with human feedback (RLHF). We explore enhancing our query producer with RAG model feedback, which is free of human efforts thus costs much less than these approaches.

## 6 Conclusion

In this work, we propose a domain adaptation framework for conversational query production. Without any human annotation in the target domain, we train a query producer using reinforcement learning with feedback from a RAG model. Besides, we also enhance our framework with some easy-to-implement techniques. Compared to fragile combinations of previous approaches, our framework is more robust against the noisy content on webpages and variations of conversations. Experiment results show that our model significantly outperforms strong baselines, especially in more challenging settings.

An important future direction can be exploring improving a query producer with LLMs. Though scaling the model size of a query producer has been demonstrated less effective (Zhou et al., 2022), we can still manage to improve a small query producer with the guidance of an LLM. Recently, researchers have greatly exploited the potential of LLMs by conducting prompt engineering, such as demonstration selection (Liu et al., 2021) or automatic prompt generation (Yang et al., 2023). We believe these work can also benefit the query production task. For example, construct pseudo training instances using an LLM to enrich the training set, or design better training signals for improving a query producer with the LLM feedback.

## Acknowledgments

The project was supported by the National Natural Science Foundation of China (No. 62276219), and the Central Leading Local Project "Fujian Mental Health Human-Computer Interaction Technology Research Center" (No. 2020L3024). We thank reviewers for their insightful comments and also thank editors for their every effort.

## Limitations

We only propose a general adaptation framework in this paper, while there are still many variants worth exploring. First, there are several studies (Shuster et al., 2021; Paranjape et al.) in these years proposed to improve the RAG model. Using these stronger models as the reward model, our model feedback can be more accurate intuitively. Second, our framework may work better in combination with other adaptation approaches such as adopting domain pretraining (Karouzos et al., 2021). However, they are both out of the scope of this paper.

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

## A  Details of Datasets

For the **source** domain, we choose Wizard-of-Internet (WoI, Komeili et al. 2022) and DuSinc (Zhou et al., 2022), both of which provide conversations annotated with search queries.

- **Wizard-of-Internet**. It is in English, built upon Bing[9] as the search engine, and is split into 35,201/2,471/2,105 training instances for training/development/test, respectively.

- **DuSinc**. It is a shared task on Chinese conversational query production containing 6,254/660 instances for training/development, respectively.[10]

For the **target** domain, we choose Wizard-of-Wikipedia (WoW, Dinan et al.), KdConv (Zhou et al., 2020), and DuConv (Wu et al., 2019). All of them are commonly used knowledge-intensive response generation datasets.

- **Wizard-of-Wikipedia**. It is an English dataset, built using Wikipedia. Following Wang et al. (2023b), we remove the first turns for all conversations, which are the keywords indicating the main topics of those sessions. We select to test on *unseen* sets, where the dialogue topics do not appear in the training set. Finally, we have 66,617/3,558/3,556 instances for training/development/testing, respectively. As it provides ground-truth documents for each instance, we can measure the quality of generated queries by comparing the retrieval results with these ground-truth documents.

- **KdConv**. It contains Chinese conversations from 3 domains (film, music, and travel). The average turn number for each utterance is much longer than DuSinc (19.0 vs. 10.2). The instances are split into 62,938/8,871/9,287 for training/development/testing. Because of lacking query annotation, we manually annotate 817 instances from the test set for query evaluation. The annotation guideline is provided in Appendix B.

- **DuConv**. It collects Chinese conversations about movies. The average character number per utterance is much shorter than that of DuSinc (10.6 vs. 21.7). We filter some unmeaningful utterances and the split for training/development/testing is 70,043/7,054/4,080. Similarly, we manually annotate 321 queries for evaluation.

[9]https://www.bing.com/
[10]We use the part that is publicly available at https://aistudio.baidu.com/aistudio/datasetdetail/139431/1

## B  Annotation Guidelines

We hire Chinese students with master's degrees and NLP backgrounds to annotate queries for KdConv and DuConv instances. Each annotator is provided with the whole dialogue session. As a query is clearer when a dialogue response is provided (usually the topic of the response), annotators are asked to label considering the information next utterance. A typical case is shown as follows:
"Context:
   知道重庆森林这部电影吗？(*Do you know the movie Chongqing Forest?*)
   知道呀，是一部由王家卫导演的片子。(*You know, it's a film directed by Wong Kar wai.*)
Response:
   而主演里更是有王菲，一上映便受到追捧。(*Faye Wong is one of the main actors. This film is highly sought after as soon as it was released.*)"

   In this case, the query should be "重庆森林演员(*actors in Chongqing Forest*)" because the response discusses one main actor "王菲(*Faye Wong*)" in it.

   If a response does not include external knowledge, we ask annotators to simply pass this case, making sure all queries are faithful to the conversations. A passed example is shown as follows:
"Context:
   这 部 电 影 有 很 多 人 是 冲 着 张 国 荣 看 的。(*Many people watch this film for Leslie Cheung.*)
   一说到他我心里就难受，怎么就那离开我们了，2003年到现在，已经离开我们十六年了。(*When it comes to him, I feel so sad. Since he passed away in 2003, there has been sixteen years.*)
Response:
   虽然离开了我们这么多年，但他给我们留下的作品，让我们永远忘不了他。(*Although he has been away from us for so many years, the works he left behind make us never forget him.*) "

## C  Input Format for *text-davinci-003*

We follow Zhao et al. (2021) to design input format for *text-davinci-003*. Besides, thanks to the advice of a reviewer, we also enrich the prompt with a task definition and speaker role information. A 2-shot case is shown as follows:

*Given the input dialogue, output a search query, which will be fed to a search engine for knowledge helpful for replying.*

*Input: "usr": "I saw a solar eclipse when i was 8 years old."*
*Output: solar eclipse*
*Input: "usr": "Hello there, are you enjoying the cold weather? I just went for a run!"*
*Output: winter runs*
*Input: "usr": "I'm trying to learn surfing at old age."*
*Output:*

Using more demonstrations usually gives better performance. We use 8 shots, which are randomly sampled from the source-domain instances.

## D  Model Configuration

We adopt T5-base / mengzi-t5-base (Zhang et al., 2021) for all experiments in English / Chinese.[11] All models are trained using AdamW optimizer (Loshchilov and Hutter) with the linear scheduler and initial learning rate of 5e-5. For source-domain query producers and RAG models, the batch sizes are set as 64 when training, and we train these models until convergence and select the checkpoints with the lowest loss values on the development set as final models. For the adaptation process, we set a larger batch size of 256 for the stability of gradient descent. Without query annotation, we stop the training until the model achieves stable average rewards of preferred queries on the development set. As for other hyperparameter selections, we empirically set $N_A$, $N$, $k$, and $\lambda$ as 10, 5, 5, and 0.1 for all experiments. For $\alpha$, $\beta$, and $\gamma$, we set 0.4 / 20, 0.2 / 5, 0.6 / 30 when using RAG / BM25 scores as feedback.

## E  Knowledge Distillation Term $\mathcal{L}_{skd}$

Both $\mathcal{L}_{kd}$ and $\mathcal{L}_{skd}$ are adopted for asking our model to learn the behavior of $\theta_{src}$, which has been well-trained on a clean labeled corpus. Thus, it can ease the impact of unexpected updating because of some noisy instances when reinforcement learning. For each input $\mathcal{X}_{<t}$, we ask our model to approximate the $\theta_{src}$ output probability distribution instead of a generated query $\hat{q}$ used in $\mathcal{L}_{kd}$:

$$\mathcal{L}_{skd} = cos\_sim(p(\hat{q} \mid \mathcal{X}_{<t}; \theta_{src}), p(\hat{q} \mid \mathcal{X}_{<t}; \theta_{tgt})), \quad (9)$$

where $cos\_sim$ is the cosine similarity function. Intuitively, probability distribution provides richer information from $\theta_{src}$. However, it can be poorly

---

[11] For all these pretrained models, we use the checkpoints from https://huggingface.co/models

| Dialogue | 不知道密云水库好不好啊，孩子需要写一篇外出游玩的文章，我正愁带他去哪儿呢。(*Is the Miyun Reservoir is a good place to visit? My child needs to write an article about going out for fun, and I'm worried about where to take him.*) 据我了解密云水库好像是自然保护区，不对外开放的。(*As far as I know, Miyun Reservoir seems to be a nature reserve and is not open to the public.*) 是吗？还好我问问你，不然就麻烦了。(*Fortunately, I asked you, otherwise it would be troublesome.*) 我记得是，你再咨询咨询。(*I remember it was. You can consult it elsewhere.*) 行，那周围那个叫田庄水库的地方呢？(*Okay, what about the place called Tianzhuang Reservoir around here?*) |
|---|---|
| Response | 我记着田庄水库不要门票，倒是挺雄伟壮观的，但是我感觉也没啥可看的。(*I remember that the Tianzhuang Reservoir doesn't require tickets, it's quite magnificent, but I don't think there's much to see either.*) |
| $f$ w/ BM25 | 密云水库的景点(*Scenic Spots of Miyun Reservoir*): 32.46 
 密云水库(*Miyun reservoir*): 33.61 
 密云水库的介绍(*Introduction to Miyun Reservoir*): **36.06** 
 田庄水库(*Tianzhuang Reservoir*): 34.16 |
| $f$ w/ RAG | 密云水库的景点(*Scenic Spots of Miyun Reservoir*): 0.06 
 密云水库(*Miyun reservoir*): 0.62 
 密云水库的介绍(*Introduction to Miyun Reservoir*): 0.38 
 田庄水库(*Tianzhuang Reservoir*): **1.44** |

Table 5: An example with their scored predictions using different $f$ from KdConv development set, where " 田庄水库(*Tianzhuang Reservoir*)" should be the target query and the key clues are underlined.

calibrated as it is not trained on the target domain thus may instead hurt model performance.

## F  Case Study for Different $f$

As shown in Table 5, We further demonstrate a case to help visualize the benefits of using RAG model feedback. The conversation first discusses about 密云水库(*Miyun reservoir*) . However, in the last turn, they start a new topic about 田庄水库(*Tianzhuang Reservoir*) . Among the 4 model predictions, only the 4-th one is correct and the others are still about the last topic. Using $f$ based on BM25, these queries are scored quite similar and the 3-rd one are scored highest. But our designed $f$ based on RAG returns more reasonable scores. The correct one is scored much higher than other queries.