# OpenReview forum: "Domain Adaptation for Conversational Query Production with the RAG Model Feedback"
_EMNLP/2023/Conference — EMNLP 2023 Findings_

### Official Review · Reviewer_HE7F · 2023-08-01

**Soundness:** 3

**Excitement:**

3: Ambivalent: It has merits (e.g., it reports state-of-the-art results, the idea is nice), but there are key weaknesses (e.g., it describes incremental work), and it can significantly benefit from another round of revision. However, I won't object to accepting it if my co-reviewers champion it.

**Missing References:**

- Chen Z, Zhao J, Fang A, et al. Reinforced question rewriting for conversational question answering[J]. arXiv preprint arXiv:2210.15777, 2022.

**Paper Topic And Main Contributions:**

This paper focuses on the problem setting of domain adaptation for conversational query generation. One of the prior works adopts reinforcement learning with nonparametric retrieval signals (i.e., BM25 score) as feedback to train query generation. This paper builds on that research by proposing a new reward function and implementing two strategies to improve stability in the RL training process. The adaptation pipeline contains three steps:
1. train a query generator on the source domain;
2. train a RAG model on the target domain;
3. train the query generator on the target domain with RL. The reward depends on the parametric retrieval score from the previously trained RAG.

The main contribution as I see is the new reward function used in RL.

**Questions For The Authors:**

**Question A:**

How does the reward normalization mentioned in Line 325 affect the results?

**Question B:**

What computational budget and computing infrastructure were used?

**Reasons To Accept:**

1. The authors propose a more accurate and effective RL feedback for query generation and make a step in developing domain adaptation methods. The method's effectiveness is encouraging because it lessens the reliance on human annotations.
2. Given that the training stability of RL is crucial, the authors do propose two simple yet effective strategies with detailed ablation studies.

**Reasons To Reject:**

**1. The proposed method needs to be clarified with more details.**

- For example, this paper uses a search engine to obtain knowledge documents (Line 143). However, it fails to provide the necessary steps for processing search results.

- In addition, this paper introduces knowledge distillation as an effective and important strategy but omits details on how to determine the pseudo query from the source generator in Line 345.

**2. The query generation evaluation on the target-domain dataset could be improved.**

- As the target-domain datasets used in this paper have no query annotation, this paper manually annotates some instances for evaluation. However, the annotations lack of reliability, which is briefly introduced without the quality control of collected data. And the word-overlap metrics between generated queries and human-written queries do not guarantee end-to-end performance (i.e., developing a better response) and therefore are not robust enough to show effectiveness.

- It would be beneficial to compare with additional RL methods. One such method, as demonstrated by Chen et al. (2022) (citation provided below), utilizes end-to-end performance as the feedback for RL.

**Reproducibility:**

3: Could reproduce the results with some difficulty. The settings of parameters are underspecified or subjectively determined; the training/evaluation data are not widely available.

**Reviewer Confidence:**

4: Quite sure. I tried to check the important points carefully. It's unlikely, though conceivable, that I missed something that should affect my ratings.

**Typos Grammar Style And Presentation Improvements:**

- The related work can be more detailed.

---

> ### Author Rebuttal · Authors · 2023-08-27
>
> 1. Necessary steps for processing search results
>
> For Wikipedia search, we follow [1] to use the summary of the wikipedia page. For Sogou search, we use the returned snippets from the search engine and clean the special tokens in them (such as email address and emoji). We will append these details and release the code.
>
> 2. Pseudo query for knowledge distillation
>
> We first follow common practice [2] to obtain pseudo query, then filter some bad ones when their query score s lower than the given threshold. We will improve this part in the revision.
>
> 3. Annotations lack of reliability
>
> As described in Appendix B, our annotators are Chinese native speakers and target queries are easier to recognize when the responses are provided. We will definitely check the annotations and also release them for future studies.
>
> 4. Evaluation metrics
>
> Uni. F1 has been adopted to evaluate query production in [3] and we additionally adopt BLEU and ROUGE to have better evaluation. Please note that we also provide the response generation results in Table 4, which also show the effectiveness of our framework.
>
> 5. End-to-end RL baselines
>
> Interesting suggestions! However, better performance might not be expected from these baselines. As shown in [4], the end-to-end model (RL-F1) does not yield better performance than other methods (RL-QR, RL-C) for QA tasks. Besides, it can be more difficult to obtain good rewards for dialogue tasks because a dialogue response is much more diverse than a question answer span. In this work, we leverage the retrieval scores from an RAG model. It is more interpretable and we also conduct instance selection based on its scores for better training.
>
> 6. Reward normalization
>
> It aims to adjust the reward within [0, 1]. Otherwise, the reward values can vary a lot across different instances, making the training unstable.
>
> 7. Computational budget
>
> All experiments are conducted on a single P40 GPU (24G).
>
> 8. Detailed related work
>
> We will improve this part. Thank you!
>
> [1] Search-Engine-augmented Dialogue Response Generation with Cheaply Supervised Query Production.
>
> [2] Understanding Knowledge Distillation in Non-autoregressive Machine Translation.
>
> [3] Link the World: Improving Open-domain Conversation with Dynamic Spatiotemporal-aware Knowledge.
>
> [4] Reinforced question rewriting for conversational question answering.

---

### Official Review · Reviewer_NqYR · 2023-08-09

**Soundness:** 3

**Excitement:**

4: Strong: This paper deepens the understanding of some phenomenon or lowers the barriers to an existing research direction.

**Missing References:**

- PK-ICR: Persona-Knowledge Interactive Context Retrieval for Grounded Dialogue https://arxiv.org/abs/2302.06674
- Reason first, then respond: Modular Generation for Knowledge-infused Dialogue https://arxiv.org/abs/2111.05204

**Paper Topic And Main Contributions:**

This paper tackles the conversational query production task, which is a task to generate search queries. It uses reinforcement learning to train the query producer model using RAG model feedback. It also explores other techniques such as knowledge distillation.

**Questions For The Authors:**

- CLEAN is only experimented with English and NOISY is only experimented with Chinese, perhaps the setting to experiment with a challenging setting in English should be suggested in the paper?

**Reasons To Accept:**

1. The paper takes care to introduce the relatively new task.
2. The paper experiments with both English and Chinese in different settings.
3. Thorough experiments are performed in regard to reinforcement learning, loss ablations, and retrieval models.

**Reasons To Reject:**

1. Figure 2 is too basic, might be good to write the model names and component names both in each case (i.e. Query Producer (T5)), and also explicitly label RL training and other processes in the figure.
2. How Question Answering domain can be adapted to dialogue tasks has been studied previously, in regards to knowledge retrieval and generation (citations below). They should be mentioned in the paper (line 049).
3.  skd-loss is only mentioned in the ablation study (section 4.3) without details. Experimental details should be included, at least in Appendix.
4. The paper should discuss more future directions on this task, as it’s very briefly discussed in Limitations and only regarding RAG models.

**Reproducibility:**

3: Could reproduce the results with some difficulty. The settings of parameters are underspecified or subjectively determined; the training/evaluation data are not widely available.

**Reviewer Confidence:**

3: Pretty sure, but there's a chance I missed something. Although I have a good feel for this area in general, I did not carefully check the paper's details, e.g., the math, experimental design, or novelty.

---

> ### Author Rebuttal · Authors · 2023-08-27
>
> 1. Figure, writings and missing citations
>
> We will definitely improve these parts. Thank you for your suggestions.
>
> 2. Challenging setting in English
>
> As commercial search engines are actually not free and we only have free access to Sogou, a Chinese one, thus we experiment on Chinese benchmarks for NOISY settings. We believe the conclusions can be the same when using different languages.

---

### Official Review · Reviewer_VdLz · 2023-08-10

**Soundness:** 3

**Excitement:**

3: Ambivalent: It has merits (e.g., it reports state-of-the-art results, the idea is nice), but there are key weaknesses (e.g., it describes incremental work), and it can significantly benefit from another round of revision. However, I won't object to accepting it if my co-reviewers champion it.

**Paper Topic And Main Contributions:**

The paper focuses on domain adaptation for the task of conversational query production, and the authors mainly follow the framework proposed by Wang et al. This study makes  improvements to two weaknesses. Firstly, they introduce a RAG model based on T5 to generate better reward socres. Secondly, a regularization term based on knowledge distillation is introduced to make the reinforcement learning training more stable. The authors have fully verified the effectiveness of the above improvements through experiments.

**Questions For The Authors:**

How does the introduction of RAG instead of BM25 affect the computational cost? Is there any quantitative comparison available?

**Reasons To Accept:**

1. The paper is complete and the motivation is clear.
2. The experiments are sufficient and effective in supporting the views of the paper.
3. The paper is well-organized and easy to read.

**Reasons To Reject:**

1. My main concern is the novelty of the paper. The authors mainly follow the architecture of Wang et al., and improve it by introducing RAG and knowledge distillation loss. These components are not new, which makes the contribution of this paper limited.

2. There is no sufficient comparison with the in-context learning method. Although the authors provided the results of text_davinci_003  in Table 2, it seems not optimal based on the prompt setting provided in Appendix C. For example, the system role was not set, that is, telling the model to complete a query generation task instead of directly giving Input-output pairs. Additionally, the authors did not use the better GPT_3.5_tubo on both CLEAN and NOISY settings. It would be interesting to test the performance of the latest LLMs on this task.

**Reproducibility:**

3: Could reproduce the results with some difficulty. The settings of parameters are underspecified or subjectively determined; the training/evaluation data are not widely available.

**Reviewer Confidence:**

4: Quite sure. I tried to check the important points carefully. It's unlikely, though conceivable, that I missed something that should affect my ratings.

---

> ### Author Rebuttal · Authors · 2023-08-27
>
> 1. Novelty of the paper
>
> Different from [1], we are the first to explore domain adaptation on this task and set up the challenging NOISY settings suffering noisy information from a commercial search engine and variation of conversations. RAG scores and knowledge distillation are adopted to tackle the problems above, which have also not been explored before. We believe our work is more practical and solid than [1].
>
> 2. More in-context learning baselines
>
> We have tested ChatGPT but it does not give more improvement (ChatGPT vs. text-davinci-003 ROUGE scores 59.45 / 49.77 / 58.22 vs. 63.19 / 48.15 / 61.99 on KdConv). We notice that it tends to reply to the dialogue instead of completing the query, though 8 demonstrations are provided. This may be because of its nature of chatting.
>
> We also test introducing speaker roles and task definition. They did not help the models based on T5-base in our early study but indeed improve the LLM peformance (ROUGE scores 67.53 / 54.23 / 66.24 on KdConv). However, it still does not outperform the basic T5-base baseline (70.95 / 59.11 / 70.08), thus our conclusions still hold.
>
> Nevertheless, leveraging LLMs to complete this task is still under-exploring. We will update these results in the revision.
>
> 3. Computational cost for the introduction of RAG instead of BM25
>
> Our framework only takes additional training costs at stage 2, where the RAG model is trained on the target domain data.  As both RAG and BM25 scores are only used for training, these models enjoy the same inference speed as the basic baseline, T5-base.
>
> [1] Search-Engine-augmented Dialogue Response Generation with Cheaply Supervised Query Production.

---

### Meta-Review · Area_Chair_J8zH · 2023-09-12

**Recommendation:** 3

**Metareview:**

The paper proposes a method for conversational query production, where search queries are generated from interlocutor utterances during a dialogue. The method builds on previous works by introducing RAG and knowledge distillation techniques.

**Pros**: After rebuttal, reviewers find the paper is well-written with most claims sufficiently defended by experiments. Most reviewers agree the paper is complete, with clear motivations and coverage of multiple languages.

**Cons**: While most concerns on soundness are addressed during rebuttal, the reviewers discussion reveals some disagreement on perceived novelty of the work. It is noted the work will be useful to the community, but in terms of methodology, the work is perceived as mostly incremental.

Given the disagreement, I took a look into these details myself. There is some lack in methodological novelty, and perhaps, more of the novelty comes from the new evaluation setup (domain adaptation) which highlights the need for the method. Still, this aspect of the paper appears somewhat underdeveloped (limited domains, no domain adaptation specific generation baselines, etc.) with most of the papers' focus on the methods.

---

### Decision · Program_Chairs · 2023-10-07

**Decision:**

Accept-Findings

**Comment:**

The paper proposes a method for conversational query production, where search queries are generated from interlocutor utterances during a dialogue. The method builds on previous works by introducing RAG and knowledge distillation techniques.

**Pros**: After rebuttal, reviewers find the paper is well-written with most claims sufficiently defended by experiments. Most reviewers agree the paper is complete, with clear motivations and coverage of multiple languages.

**Cons**: While most concerns on soundness are addressed during rebuttal, the reviewers discussion reveals some disagreement on perceived novelty of the work. It is noted the work will be useful to the community, but in terms of methodology, the work is perceived as mostly incremental.

Given the disagreement, I took a look into these details myself. There is some lack in methodological novelty, and perhaps, more of the novelty comes from the new evaluation setup (domain adaptation) which highlights the need for the method. Still, this aspect of the paper appears somewhat underdeveloped (limited domains, no domain adaptation specific generation baselines, etc.) with most of the papers' focus on the methods.